# Thermally enhanced photoluminescence for heat harvesting in photovoltaics

Assaf Manor[1], Nimrod Kruger[2], Tamilarasan Sabapathy[3] & Carmel Rotschild[1,2,3]

The maximal Shockley–Queisser efficiency limit of 41% for single-junction photovoltaics is primarily caused by heat dissipation following energetic-photon absorption. Solar-thermophotovoltaics concepts attempt to harvest this heat loss, but the required high temperatures ($T > 2,000$ K) hinder device realization. Conversely, we have recently demonstrated how thermally enhanced photoluminescence is an efficient optical heat-pump that operates in comparably low temperatures. Here we theoretically and experimentally demonstrate such a thermally enhanced photoluminescence based solar-energy converter. Here heat is harvested by a low bandgap photoluminescent absorber that emits thermally enhanced photoluminescence towards a higher bandgap photovoltaic cell, resulting in a maximum theoretical efficiency of 70% at a temperature of 1,140 K. We experimentally demonstrate the key feature of sub-bandgap photon thermal upconversion with an efficiency of 1.4% at only 600 K. Experiments on white light excitation of a tailored Cr:Nd:Yb glass absorber suggest that conversion efficiencies as high as 48% at 1,500 K are in reach.

[1] Russell Berrie Nanotechnology Institute, Technion—Israel Institute of Technology, Haifa 32000, Israel. [2] Grand Energy Program, Technion—Israel Institute of Technology, Haifa 32000, Israel. [3] Department of Mechanical Engineering, Technion—Israel Institute of Technology, Haifa 32000, Israel. Correspondence and requests for materials should be addressed to C.R. (email: Carmelr@tx.technion.ac.il).

Single-junction photovoltaic (PV) cells are limited in efficiency by the Shockley–Queisser (SQ) limit[1] of ∼33% at the one-sun illumination level (1,000 Wm⁻²) and 41% at the maximum solar illumination level. For PVs with relatively low bandgaps (for example, $E_g = 1$–1.4 eV), efficiency loss is primarily caused by heat dissipation during the process of electro-chemical potential generation[2]. Several approaches have been suggested to eliminate this heat loss via different physical mechanisms, including down-conversion of high-energy photons[3], multiple-exciton generation[4], hot-carrier cells[5] and multi-junction PVs. To date, only multi-junction PVs have achieved ultra-high efficiencies near 46% (ref. 6), but they remain a rather complex solution involving the fabrication of dozens of hetero-layers in a single device[7]. A different approach is to treat solar irradiation not only as a photon source but also as a heat source. In photo-thermal concepts, such as solar thermo-photovoltaics (STPV)[8,9], solar heat flux is first converted to a thermal emission by a selective absorber/emitter and is then absorbed by a low bandgap-matching PV cell. Although the projected conversion efficiencies of these systems are impressive, high absorber temperatures above 2,000 K (ref. 10) are required to overcome the SQ limit because only the energetic portion of the thermal radiation is harvested. After over thirty years of research, the record conversion efficiency for STPV stands at 3.2% for an absorber operating temperature of 1,285 K (ref. 11).

Alternatively, if light is absorbed by a photoluminescent (PL) material, both photonic and thermal excitations are generated. Under such excitation, the material's PL is described by the generalized Planck's law[12,13]:

$$R(\hbar\omega, T, \mu) = \varepsilon(\hbar\omega) \cdot \frac{(\hbar\omega)^2}{4\pi^2\hbar^3c^2}\frac{1}{e^{\frac{\hbar\omega - \mu}{k_BT}} - 1}$$

$$\cong R_0(\hbar\omega, T) \cdot e^{\frac{\mu}{k_BT}} \qquad (1)$$

where $R$ is the emitted photon flux in photons per second per unit area and solid angle, per energy interval. In this study, $T$ is the temperature, $\varepsilon$ is the emissivity, $\hbar\omega$ is the photon energy, $c$ is the speed of light in vacuum and $k_B$ is Boltzmann's constant. The full term can be approximated by the multiplication of the thermal emission rate $R_0(\hbar\omega, T)$ with the exponential term containing the chemical potential $\mu$, which describes the excitation above thermal equilibrium. We have recently investigated the interplay of heat and PL in generating thermally enhanced PL[14] (TEPL), where PL was shown to be able to extract thermal energy with minimal entropy generation by thermally induced blueshift of a conserved PL rate. This ability is manifested by the TEPL's enhanced energetic photon rates in

comparison to the equal-temperature thermal emitter. Therefore, it is constructive to replace the traditional STPV thermal absorber with a TEPL absorber, where the energetic photons are harvested by a high bandgap solar cell leading to high device efficiencies via voltage enhancement above the SQ limit.

We start with the device thermodynamics and its efficiency analysis. Next, we experimentally demonstrate a key feature of a TEPL device, in which $\lambda = 914$ nm photons (sub-bandgap to the GaAs PV at 850 nm) are thermally upconverted by a glass:Nd³⁺ absorber to wavelengths shorter than 850 nm and are harvested by a GaAs solar cell with an efficiency of 1.4% at 600 K. We then continue to white-light demonstration, where a tailored Cr:Nd:Yb absorber is shown to exhibit broad-band TEPL conversion of sub-bandgap photons (850 nm < $\lambda$ < 1,100 nm) to photons accessible to GaAs PV. Based on the conversion results, we simulate a practical TEPL device that yields maximal practical efficiencies of 48% at operating temperature of 1,500 K.

## Results

**Theoretical TEPL model.** Based on our previous study[14] we first consider the comparison between TEPL and thermal emitters at elevated temperatures. Figure 1a depicts the evolution of the PL spectrum emitted from a 1.1 eV bandgap material. With temperature increase, the rate of emitted photons is conserved and the spectrum is blue-shifted towards higher energies. This process continues as long as the chemical potential is positive (Fig. 1b, dotted line). When the chemical potential vanishes at $T > 1,200$ K, the emission becomes thermal and the number of emitted photons increases (shown by the red spectrum at 1,300 K). Figure 1b also depicts the high-energy photon rate ($E_{photon} > 1.45$ eV) compared with that of thermal emission at the same temperatures. Evidently, the thermal rate is orders of magnitude lower, and, therefore, not useful in the relatively low temperature regime ($T < 1,000$ K).

For the thermodynamic analysis, we consider a theoretical TEPL device consisting of a thermally insulated, low bandgap TEPL absorber that completely absorbs the solar spectrum above its bandgap ($E_{g,Abs}$) as depicted in Fig. 2a. Energetic photon absorption increases the absorber's temperature by electron thermalization, and induces thermal upconversion of cold electron-hole pairs, as indicated by the arrows. The resulting emission spectrum is TEPL, which, according to equation (1), is described by $T_{high}$ and $\mu_{TEPL} > 0$.

While the thermally upconverted portion of the TEPL above the $E_{g,PV}$ bandgap is harvested by a room-temperature PV, sub-bandgap photons are reflected back to the absorber by the PV cell back reflector, as in state-of-the-art GaAs cells[15,16]

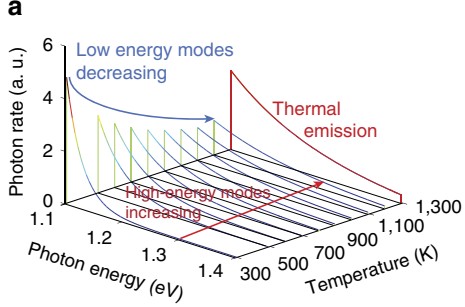
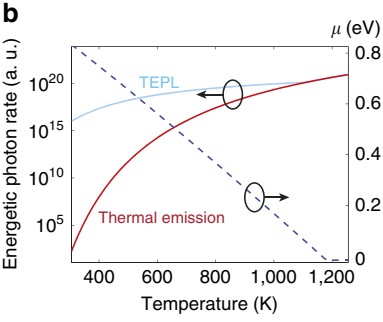

**Figure 1 | The TEPL effect. (a)** The evolution of PL spectrum emitted from a 1.1 eV bandgap material as a function of temperature. The blue arrow marks the decrease in the low-energy photon rates while the red arrow marks the increase in the high-energy photon rates. This process continues as long as $\mu > 0$ (also seen by the dotted line in **b**). The last spectrum (in red) is thermal emission, which dominates after $\mu$ vanishes. **(b)** A comparison between the rate of energetic photons ($E_{Photon} > 1.45$ eV) in TEPL (blue) and thermal emission (red), as a function of temperature. The chemical potential is shown by the dotted blue line. It decreases monotonically with temperature and vanishes near 1,200 K.

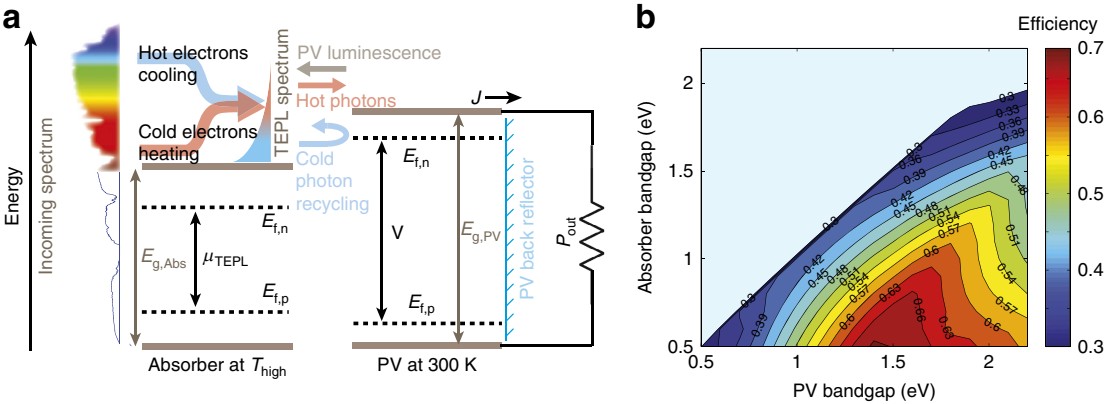

**Figure 2 | Energy conversion dynamics and ideal efficiency. (a)** The TEPL conversion dynamics. Solar spectrum above $E_{g,Abs}$ is absorbed by the luminescent absorber and emitted as TEPL towards the PV. Sub-bandgap photons are recycled back to the absorber (blue arrow) while above $E_{g,PV}$ photons are converted to current. For an ideal PV, its PL is also recycled to the absorber (grey arrow). **(b)** Ideal system efficiency as a function of the absorber and PV bandgaps.

(blue arrow in Fig. 2a), maintaining the high TEPL chemical potential. The emitted PV luminescence, which in the radiative limit has an external quantum efficiency (EQE) of unity, is also recycled back to the absorber (grey arrow). Thus, the otherwise dissipated thermalization energy of the absorber is converted to increased voltage and efficiency at the PV. The ability to generate both high current (due to the absorber low bandgap) and high voltage paves the way to exceeding the SQ limit, inherently set by the single-junction PV current-voltage tradeoff.

The device thermodynamic simulation is achieved by detailed balance of photon fluxes, based on equation (1). The calculation accounts for the different systems variables, such as the two bandgaps, the solar concentration ratio upon the absorber, the absorber's EQE, the sub-band photons recycling efficiency (PR) and the PL EQE of the PV. The simulation yields the device's I–V curve at various operating temperatures, from which the system's efficiency can be deduced (see Supplementary Note 1).

The simulation results of the maximal theoretical efficiency for each absorber and PV bandgap combination, when all the parameters are set to their ideal values are depicted in Fig. 2b. For each $E_{g,Abs}$, the efficiency initially increases with the increase in $E_{g,PV}$, but decreases for higher values due to the tradeoff between voltage gain at the PV and loss of photons due to the reduction in the harvested portion of the spectrum. This tradeoff sets a maximal efficiency of 70% for $E_{g,Abs} = 0.5$ eV and $E_{g,PV} = 1.4$ eV, at a temperature of 1,140 K. Since the bandgaps must correspond to available absorber materials and PV technologies, we proceed with a specific absorber and PV combination of $E_{g,Abs} = 1.1$ eV and $E_{g,PV} = 1.45$ eV, which, as will be experimentally shown, correspond to a tailored rare-earth doped absorber matched to GaAs PV cell. Figure 3a shows the efficiency of such bandgap combination as a function of temperature, for several solar concentration levels while the absorber EQE and PR are set to unity. For one-sun irradiance, two cases are plotted. When the PV cell PL EQE is set to unity, efficiencies above 50% are obtained (blue curve). Interestingly, in this regime it is possible to work at low temperatures (500 K) with 50% efficiency, or at moderate temperatures (1,000–1,300 K), which raise the efficiency to above 57%. When setting the PV PL EQE to 24.5% (highest reported value for GaAs cells[17]) the curve decreases to a maximal efficiency of 48%, which is obtainable at temperatures higher than 600 K (black curve). It may seem odd that such temperatures are achievable with no solar concentration. However, this is a result of the ideal thermal insulation and photon-recycling. Further quantitative analysis of

the efficiency dependence on realistic values of the absorber's EQE and PR values will be presented at the discussion, where we simulate the efficiency of a practical device. For the 24.5% PV PL EQE case at higher solar concentration levels, the curves are shifted towards higher temperatures and efficiencies above 50% at temperature range 1,000–1,500 K can be obtained. For the unity PV PL EQE case, solar concentration enhance efficiency very little (not shown on graph).

For all the cases, the resulted efficiency from a thermal emitter (STPV) operating at the same temperatures is shown by the dotted red curves. The TEPL and STPV curves merge at high temperatures, as also shown in Fig. 1b. Although thermal emission achieves equal efficiencies to TEPL in the relatively high temperature regime, it rapidly decreases with temperature whereas the TEPL's efficiency plateau achieves much greater efficiencies at the moderate temperature regime.

In order to deepen our understanding on the origin of the efficiency increase, we plot the system's I–V curves for the one-sun irradiation case in Fig. 3b. For comparison, we also plot a 1.1 eV bandgap, SQ limited PV. It is seen that the while the system's current is identical to the SQ cell, the open-circuit voltage ($V_{oc}$) increase is responsible for the efficiency enhancement, in the two TEPL cases. The 100% PV EQE case shows the highest efficiency due to the recycling of the PV luminescence by the absorber, causing its chemical potential to increase relative to the 24.5% PV EQE case. This is shown by the two curves at the inset, depicting the absorber's chemical-potential temperature dependence.

**Towards a TEPL device realization.** Next, we investigate the realization of a practical TEPL device based on the above-mentioned bandgap combination of $E_{g,Abs} = 1.1$ eV (1,100 nm) and $E_{g,PV} = 1.45$ eV (850 nm). First, the absorber's EQE is essential for building up the chemical potential[14] and hence for the system's efficiency. Solid state semiconductors, such as GaAs, excel in EQE at room temperature, but the EQE decreases markedly with temperature due to non-radiative recombination mechanisms[18]. Additionally, EQE requires photon extraction, which creates an additional challenge to semiconductors because of their high refractive index. Conversely, rare-earth ions, such as neodymium and ytterbium, provide excellent performance because their electrons are localized and insulated from interactions[19]; this results in the conservation of their high quantum yields at extremely high temperatures, as we have recently shown in a silica:$Nd^{3+}$ system[14]. Based on these advantages, our experimental

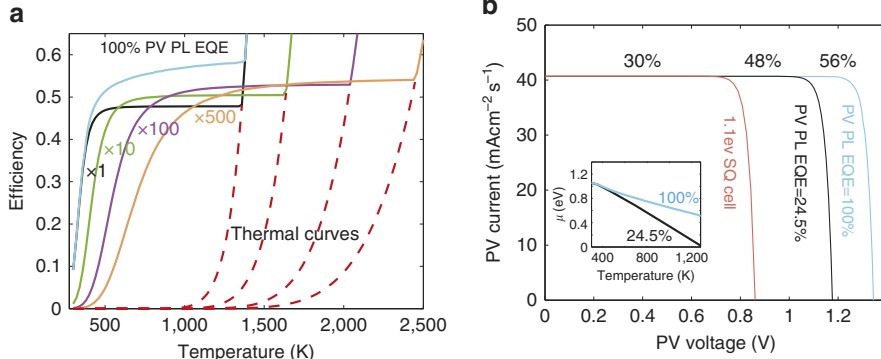

**Figure 3 | TEPL device thermodynamic analysis.** (**a**) Efficiency-temperature dependence, with ideal parameter settings. The different concentration levels are marked by *X*, which correspond to the different curves by their colour. For one-sun illumination, the blue curve depicts the efficiency when the PV PL EQE is unity, while for the rest it is set to 24.5%. The red curves shows the system efficiency if the absorber is replaced with a thermal emitter; each curve merges with a TEPL curve of the same solar concentration. (**b**) *I–V* curves of the two TEPL systems (24.5 and 100% PV PL EQE), at one-sun illumination, in comparison to a SQ limited cell of 1.1 eV bandgap. The inset shows that the voltage enhancement originates from the absorber's increased chemical potential.

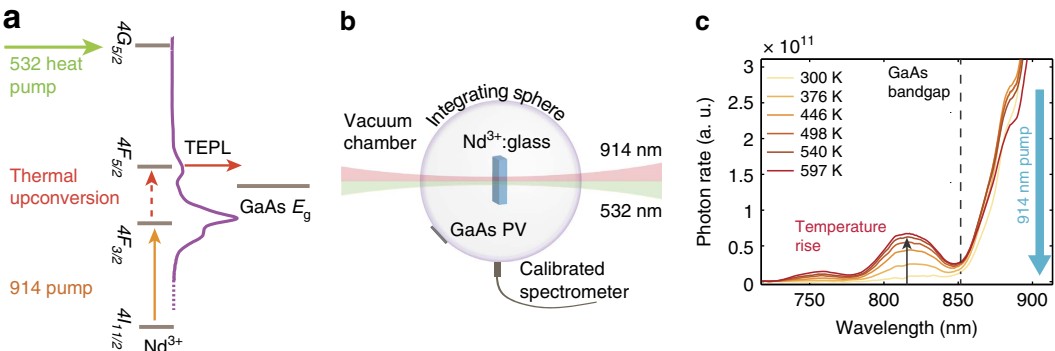

**Figure 4 | Experimental TEPL demonstration.** (**a**) TEPL upconversion energy diagram. The $Nd^{3+}$ system is pumped by a sub-bandgap source of $\lambda = 914$ nm (relative to the GaAs cell, orange arrow) for PL generation at the $4F_{3/2}$ level. The system is also pumped by 532 nm photons that are absorbed by the $4G_{5/2}$ level, followed by fast thermalization heating the sample. Thermally upconverted photons emitted from the $4F_{5/2}$ level are marked as TEPL, and are harvested by the GaAs cell. (**b**) Experimental set-up of the $Nd^{3+}$: glass sample in an integration sphere in vacuum, co-pumped by the 532 and 914 nm lasers. (**c**) Temperature dependence of the PL spectrum. The 914 nm pump is shown on the right (blue arrow), and the GaAs bandgap is shown by the dotted line. Upon temperature rise, the TEPL spectrum is enhanced.

demonstration begins with quantitative measurements of the thermal upconversion efficiency of sub bandgap photons at 914 nm, which is the TEPL's key feature. For this monochromatic excitation experiment, we use an off-the-shelf $Nd^{3+}$ glass. However, because the $Nd^{3+}$ absorption spectrum is discrete and its absorption coefficients are low (typically $5–10$ cm$^{-1}$), it must be sensitized to operate under white-light excitation. For a practical TEPL converter, we follow with experimental demonstration of broadband sensitization and upconversion at high temperatures.

**Monochromatic TEPL upconversion.** The upconverting experiment is described in Fig. 4a. We use a sub-bandgap photon source of 914 nm (1.35 eV), which is absorbed by the $Nd^{3+}$ $4F_{3/2}$ level, for the photo-excitation and generation of the chemical potential of this level. This light source cannot contribute to the GaAs cell photocurrent, thus making any observed photocurrent induced only by the thermal effect. The heat source is a 532-nm laser pump, which is absorbed by the $4G_{5/2}$ and $2G_{7/2}$ levels. Photon absorption is followed by fast thermalization of electrons to lower energy levels, leading to the sample heating and subsequent thermal upconversion of electrons from the $4F_{3/2}$ level to the $4F_{5/2}$ level, followed by emission of upconverted photons above the

GaAs cell bandgap. Figure 4b depicts the experimental set-up. The sample is vacuum-insulated, in an integration sphere. The sample's emission is shone on the GaAs PV cell and is measured by a calibrated spectrometer (see Methods section).

First, we turn on the 532-nm laser until the sample's temperature reaches a steady state. Then, the laser is switched off, and we monitored a negligible current at the PV, which indicates on the sample's negligible thermal emission. This step is critical because the 532-nm pump induces PL at the probed $4F_{3/2}$ level in addition to the heating effect. Before switching on the 914-nm pump we wait for 1 s, much longer than the PL lifetime of $\sim 300$ µsec (http://www.schott.com/advanced_optics). This ensures that the 532 nm induced PL vanishes. Due to the vacuum thermal insulation, the sample maintains its high temperature upon the introduction of the 914 nm pump. Figure 4c shows the resulting TEPL power spectrum, which is enhanced upon temperature increase. Figure 5a shows the corresponding evolution in the PV *I–V* curves (produced only by the thermally induced blue-shifted 914-nm pump). Figure 5b shows the upconversion efficiency versus temperature. The red curve shows the upconversion efficiency of the photons from the 914 nm pump to energies above the GaAs bandgap of 1.45 eV ($\lambda < 850$ nm), as extracted from the power spectrum. Although it

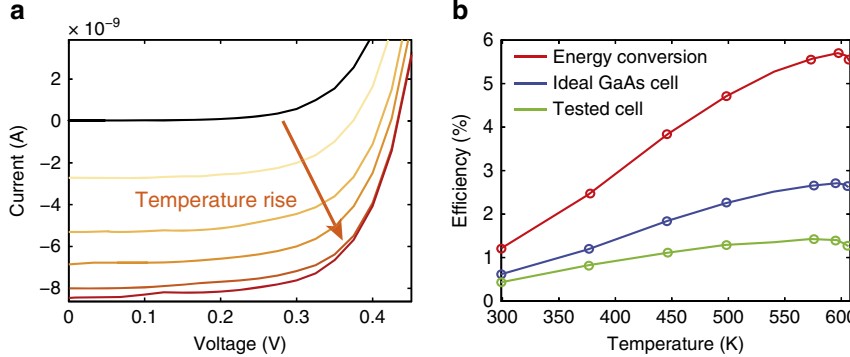

**Figure 5 | TEPL upconversion. (a)** The GaAs PV cell *I–V* curves evolution with temperature. **(b)** Total energy upconversion efficiency (red), ideal cell projected upconversion efficiency (blue) and tested PV efficiency (green) dependence on temperature.

peaks at ~6%, the real PV conversion efficiencies are lower because of the limited PV efficiency. The green plot shows the conversion efficiency of the 1-mm² cell (which was measured to have $\eta = 17\%$ under 1 Sun illumination). Because this cell is far from ideal, the blue line in Fig. 5b shows the calculated efficiency of a state-of-the-art GaAs cell[15] ($\eta = 28.8\%$ under 1 Sun illumination). Noticeably, even at room temperature a relatively small blue shift is detected, resulting in a 0.3–0.5% efficiency for the tested and state-of-the-art cells. The efficiency peaks at 600 K, with values of 1.4% and 2.5% for the tested and state-of-the-art cell, respectively. As shown, efficiencies deteriorated above 600 K. Such a limit was not observed when rare-earth materials were doped in pure silica, in which temperatures above 1,300 K have been reported[14]. We thus hypothesize that the silicate glass used for this experiment exhibits thermal degradation of the PL at high temperatures. Extrapolating the experimental results for an Nd:SiO₂ system up to 1,000 K would reach ~6.7% (see Supplementary Note 2 and Supplementary Fig. 1). In addition, we have measured the sample's EQE to be ~30%, whereas Nd³⁺ doped glasses are reported to achieve PL EQE's of almost 100% (refs 20–22) at relatively low Nd³⁺ concentration levels of 1 wt%.

**Broadband TEPL upconversion.** We now proceed from monochromatic upconversion to broadband excitation of TEPL. Since an Nd³⁺ doped absorber is not suitable for white-light harvesting, we fabricated a tailored Cr:Nd:Yb glass absorber (doping concentrations: 0.1:1:0.4 wt%, see preparation details at the methods section), which continuously absorbs sunlight in the visible–near-infrared (NIR) part of the spectrum (400–1,100 nm; see Supplementary Fig. 2 with the association of different spectral bands to the absorbing dopant). The Cr³⁺ dopant serves as an efficient sensitizer of Nd³⁺ in the 300–600 nm regime[23,24], and the Yb³⁺ completes the lacking Nd³⁺ absorption near 1,000 nm, in order to form a continuous NIR band. The room temperature emission of this at 850–1,100 nm will be shifted by the TEPL towards $\lambda < 850$ nm. At room temperature, the Nd³⁺ 1,064 nm line does not absorb light, due to its 4-level characteristics. However, at high temperatures the lower level of this transition ($^4I_{11/2}$) is thermally populated by the ground level. At 1,500 K, about 20% of the electrons occupy this level, giving rise to light absorption. Before full white-light excitation, we verify TEPL conversion of this material composition at the 900 nm (Nd³⁺), 980 nm (Yb³⁺), and 1,064 nm (Nd³⁺) bands composing the sub-band regime (850 nm < $\lambda$ < 1,000 nm). Here, in addition to the pumps, which do not heat the sample due to lack of thermalization, we use a CO₂ laser to raise the temperature. Figure 6a depicts this upconversion experiment. The three monochromatic pumps are shown both on the spectrum and at

the energy diagram below. The green arrows indicate the TEPL conversion from each of the bands. We qualitatively validate the TEPL conversion by monitoring the spectral evolution with the rise in temperature. The spectral data for the 1,064 nm pump is presented in Fig. 6b, showing the rise in TEPL emission upon the sample's heating, further validating our claim regarding the 1,064 nm light absorption. We verify that the signal is not thermal emission by switching off the NIR pump and monitoring negligible signal under CO₂ excitation. Similar conversion was observed under the 900 nm excitation as previously shown for Nd³⁺ alone. The TEPL conversion was also observed for the 980 nm pump, indicating sensitization from the Yb³⁺ $^2F_{5/2}$ level to the Nd³⁺ $^4F_{3/2}$ level, as shown at the energy diagram (please refer to the Supplementary Note 3 and Supplementary Figs 3 and 4 for the 900 and 980 nm pump spectral data and for the quantitative assessment of the 1,064 nm line absorption coefficient).

Next, we pump the sample with white light and measure the overall TEPL effect. For this purpose, we use a supercontinuum white laser source, with a continuous lineshape across the sample's absorption spectrum (see Supplementary Fig. 2). Figure 6c shows the evolution of the sample's PL with the heating, resulting from the electron thermalization following white light absorption: While the sub-bandgap emission ($\lambda > 850$ nm, marked by a blue arrow) decreases, the TEPL (marked by an orange arrow) is increased. It is not surprising that this evolution is similar to the monochromatic excitation case (Fig. 4b), as the PL spectrum only depends on the rate of absorbed/emitted photons (chemical potential) and the temperature. We have reached a maximal temperature of 650 K with the supercontinuum pump (we relate this limit to the low content of high-energy photons in the pump, as seen in Supplementary Fig. 2). Since the device's operating temperature range is higher, we used an additional CO₂ laser as a heat source in order to further raise the sample's temperature above 650 K. As a measure of the TEPL's upconversion of photons into the $\lambda < 850$ nm band, Fig. 6d shows the monotonically rising ratio between the rate of photons below 850 nm (above 1.45 eV) to the total rate of emitted photons. This ratio reaches nearly 30% at 1000K. The dotted line marks the temperature from where the CO₂ laser is used. In the next section, we use this data for the simulation of a practical device, in which the sample's TEPL is coupled to GaAs PV cells.

## Discussion

The TEPL upconversion efficiency may be compared with other upconversion mechanisms relevant to incoherent radiation, such as triplet-triplet annihilation[25], rare-earth upconverters[26–28]

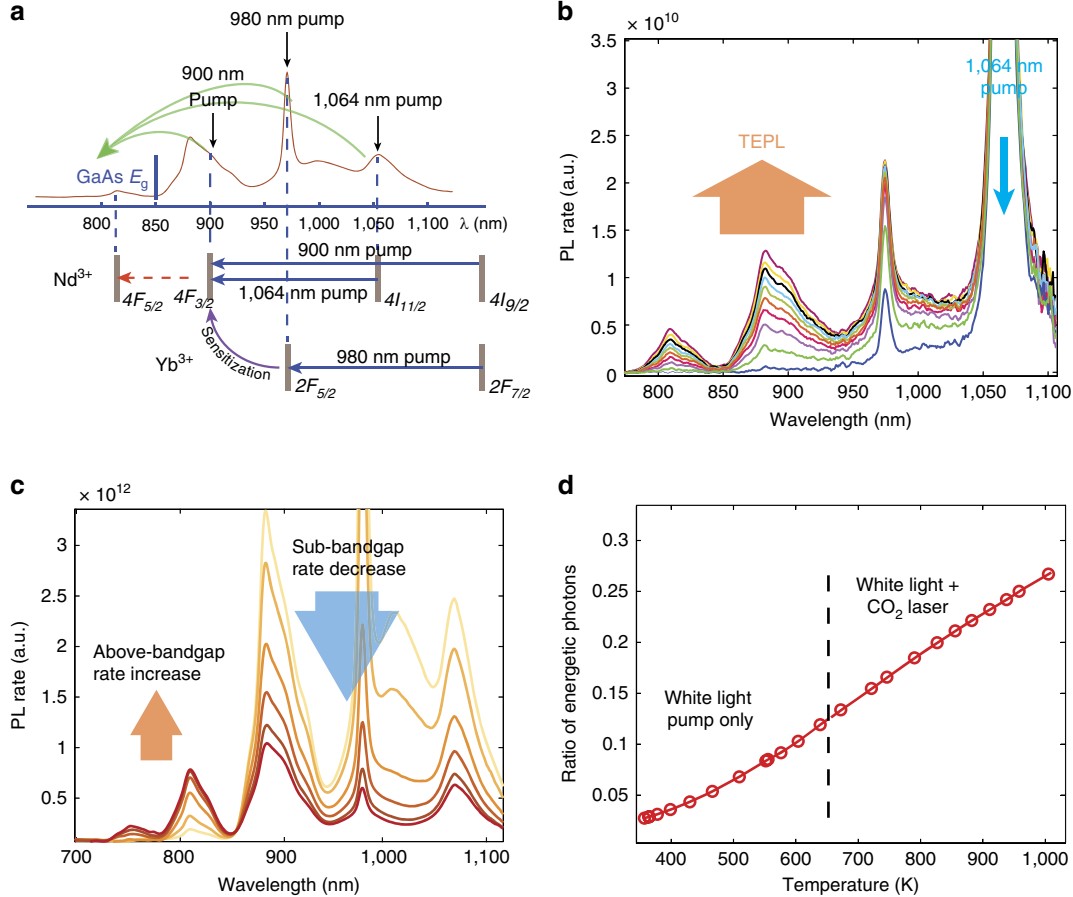

**Figure 6 | Broad-band TEPL upconversion.** (**a**) The Cr:Nd:Yb glass sample lineshape in the NIR. The three pumps used are indicated with their corresponding electronic transitions in blue arrows (two for the $Nd^{3+}$ and one for the $Yb^{3+}$). The purple arrow indicates energy transfer between the $Yb^{3+}$ and the $Nd^{3+}$. The red dotted line indicates the thermal upconversion process, resulting in TEPL emission. (**b**) The rise of TEPL emission (marked by an orange arrow) for the 1,064 nm pump (blue arrow), upon heating. (**c**) The PL spectrum evolution showing the reduction of sub-bandgap photon rate (blue arrow, $\lambda > 850$ nm) and the rise in the rate of photons of $\lambda < 850$ nm (orange arrow) under white-light excitation. (**d**) Temperature dependence of the ratio of energetic photon ($\lambda < 850$ nm) to total photon rates, for white-light pumping (supercontinuum laser). The dotted line marks the temperature where a $CO_2$ laser was used as an additional heat source.

and quantum dots[29]. Under moderate light intensities, these methods exhibit best efficiencies of 3–5% (refs 25,30,31), which are comparable to the demonstrated upconversion efficiency. Particularly, ref. 28 demonstrated record efficiency thermal upconversion of 16%. However, this was achieved in ultra-high temperatures of almost 3,000 K, exactly supporting our main claim: the same thermal emission at sample temperature of 600 K would yield negligible efficiency of $\eta = 10^{-8}$ due to the missing chemical potential counterpart.

Encouraged by these results, we continue with simulating a practical white light device based on a TEPL absorber and GaAs PV. Our device is an elongated cylindrical absorber illuminated by concentrated sunlight through its narrow facet (Fig. 7a). Since the absorber's absorption coefficient is relatively low ($\alpha < 10$ cm$^{-1}$), it is possible to spatially separate the solar absorption and the PL emission. While sunlight is guided along the absorber, the emitted PL is efficiently extracted to the periphery (normal to the optical axis) as long as the cylinder diameter is smaller than the material's absorption length ($D < 1/\alpha$).

The entire periphery of the absorber is laid with GaAs PV cells, which, as in the conceptual model, reflect sub-bandgap photons back to the absorber. We use the presented capability of the Cr:Nd:Yb absorber to upconvert photons from the NIR band to

wavelengths below 850 nm, and simulate the performance of such a device; considering the absorber's temperature, EQE, PR efficiency and solar concentration (please refer to the Supplementary Note 4 for simulation details).

While the system results, presented in Fig. 3, were analysed for ideal parameters (that is, EQE = 100% and PR = 100%), for the practical analysis one has to choose realistic values. Figure 7b shows the system's efficiency at a maximal operating temperature of 1,500 K as a function of the solar concentration. At this stage we simulate the conversion efficiency for non-ideal EQE and PR values, both equal 0.9, 0.95 and 0.98 (green, blue and purple lines, respectively).

For every case, the option of PV PL EQE = 24.5% (as for state-of-the-art cells, marked in rectangles) and PV PL EQE = 100% (marked in circles) was examined. In addition, the SQ limit of a 1.1 eV bandgap cell, similar to the TEPL absorber bandgap, is plotted for comparison. We divide the efficiency to three main ranges, as seen by the dotted horizontal lines at 41 and 45%. Evidently, the SQ limit is exceeded through all concentration levels. However, if we compare the system's efficiency to efficiencies above ~41% (the maximal SQ limit), we notice that either high concentrations ($C > 1,000$ suns) are needed for the lowest PR and EQE, or moderate concentrations ($C > 10$ suns) for higher PR and EQE (above 0.9).

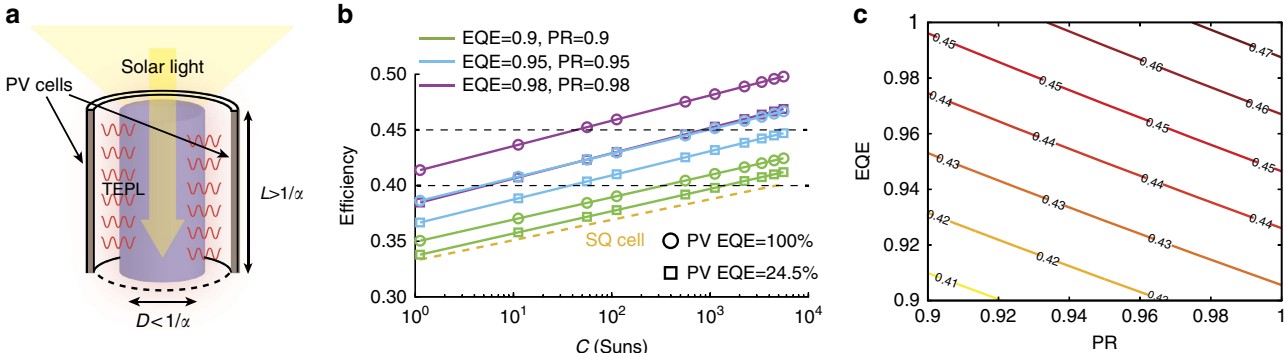

**Figure 7 | A TEPL practical device. (a)** Device model. Solar light is concentrated on the upper absorber facet and absorbed in the long dimension $L$, while PL photons are extracted through the short dimension $D$ towards the peripheral PV cells. **(b)** Projected efficiencies as a function of the solar concentration. Three values of both EQE and PR were chosen: 0.9 (green), 0.95 (blue) and 0.98 (purple). In addition, each curve is calculated for PV PLEQE = 24.5% (square symbols) and PV PLEQE = 100% (circle symbols). The SQ limit is plotted by the dotted yellow curve. The two dashed horizontal lines divide the efficiency to three ranges: low (35–40%), high (40–45%) and ultra-high (45–50%). **(c)** Efficiency dependence on PR and EQE at $C = 3,000$ suns. The efficiency span of 41–48% is achievable for EQE and PR values above 0.9.

Ultra-high efficiencies above 45% are enabled for the highest combination of PR and EQE (at $C > 100$ suns) or for the ideal case of PV PL EQE = 100% with PR, EQE = 0.95. We see that, as expected, the PR and EQE values have a critical influence on the system's efficiency, and high values ($> 0.9$) are required in any scenario. A better examination of these parameters is given in Fig. 7c, showing the efficiency dependence on any PR and EQE value in this range at $C = 3,000$ suns, operating temperature of 1,500 K, and PV PL EQE of 24.5%. The entire span of efficiencies 40–48% is covered by these values.

It is important to note that the required EQE and PR are in reach. The EQE is a product of the internal QE for each of the emitting dopant elements ($Nd^{3+}$ and $Yb^{3+}$) and the photon extraction efficiency. As previously mentioned, extremely high EQE values were reported for these Rare-Earth dopants. For $Nd^{3+}$, EQE's above 95% were reported for $Nd^{3+}$ concentrations of 1 wt% or lower[20–22], which is also the $Nd^{3+}$ concentration level of our tailored absorber. For $Yb^{3+}$, the science of optical cooling of solids pushed the limits of EQE to unprecedented values above 99% (refs 32,33) due to the need to efficiently emit heat-bearing PL photons, similarly to the TEPL requirement. The required high PR efficiency sets an additional engineering challenge. State-of-the-art cells exhibit 98% reflectivity for sub-bandgap photons[34]. Multiple reflections in the device would ensure the condition for re-absorption of these photons in the absorber, if the photon travels at least the materials absorption length $L = 1/\alpha$. For an absorber of diameter $L/3$ with 98% reflectivity back-reflectors, the PR is $0.98^3 \cong 0.95$, matching the efficiency range of 40–45% (for EQE $> 0.9$).

In order to get to the PR regime of 0.95–0.99, which is required for ultra-high efficiencies, reflectivity values of $R = 98$–99.5% are needed. Such (and higher) reflectivity values are obtainable with current back-reflector and filter technologies, even for omnidirectional purposes such as ours[35–38].

To conclude, in this paper we analyse and demonstrate the concept of a TEPL based solar energy converter, which can exceed the SQ limit by utilizing PL as an optical heat pump upconverting sub-bandgap photons. We show that, in contrast to the traditional STPV concept requiring extremely high working temperatures, a TEPL converter can ideally achieve efficiencies of up to 70% at moderate temperatures of 1,000–1,500 K. Experimentally, we demonstrate a key feature of a TEPL device, in which 914 nm sub-bandgap photons are thermally blue-shifted

and harvested by a GaAs solar cell with an efficiency of 1.4% at only 600 K. We then continue to white-light demonstration where a tailored Cr:Nd:Yb glass absorber exhibits broadband thermal upconversion of photons from the sub-bandgap spectral interval 850 nm $< \lambda <$ 1,100 nm to $\lambda <$ 850 nm. Based on these results, we simulate a practical TEPL device where the absorber is coupled to GaAs PV cells, and show that efficiencies of 45–50% are in reach.

## Methods

**Upconversion experiment.** For the experimental set-up, a 10 mm × 2 mm × 2 mm $Nd^{3+}$: silicate glass (Schott LG-680) sample was held in an integration sphere under vacuum conditions ($10^{-4}$ mbar). The PL sample was first heated by a high-intensity 532-nm laser (Coherent Verdi V6). Then, a 914-nm laser pump (CNI lasers, 1 W with an estimated absorbed fraction of 16%) was turned on 1 s after the 532-nm pump was turned off. The sample then emitted TEPL due to the 914-nm pump and the 532-nm residue heat. Placed inside an integrating sphere, the sample emission was shone on a fiber coupled spectrometer (Ocean optics QE 65000) to measure the absolute irradiance and on a 1 mm² GaAs PV cell to measure the power via a source meter (Keithley 2400). The system was calibrated by illuminating the integrating sphere with an irradiance standard lamp (Newport). The efficiency was calculated by dividing the PV electrical power by the total absorbed 914-nm power, which was measured with the integrating sphere. The temperature of the sample was measured at each step via fluorescence intensity ratio thermometry[14,39] (see Supplementary Note 5 for details).

**Broadband converter glass preparation.** Silica glass matrix is synthesized by melt quenching technique. Appropriate quantities of oxide materials (silica, sodium oxide and calcium oxides) with the rare earth elements and transition metal (neodymium oxide, ytterbium oxide and chromium oxide) are put into an alumina crucible. It is subsequently heated in a box furnace to a temperature above the melting point of the constituents at a heating rate of 10 °C min⁻¹. The crucible containing the melt is mixed a couple of times to ensure homogeneity. It is afterwards quenched in air to obtain bulk glasses. For the TEPL experiment, Flat fibers are drawn from the melt in order to obtain appropriate size samples.

**Data availability.** The data that support the findings of this study are available from the corresponding author upon request.

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

## Acknowledgements

The research leading to these results was supported financially by the European Union's Seventh Framework Programme (H2020/2014–2020]) under grant agreement no. 638133-ERC-ThforPV. This report was partially supported by the Russell Berrie Nanotechnology Institute (RBNI) and the Grand Technion Energy Program (GTEP) and is part of The M. Leona and B. Harry. Helmsley Charitable Trust reports on the Alternative Energy series of the Technion and the Weizmann Institute of Science. We would also like to acknowledge the partial support provided by the Focal Technology Area on Nanophotonics for Detection. A. Manor would like to thank the Adams Fellowship program for financial support, and Prof C. Rotschild would like to thank the Marie Curie European Reintegration Grant for its support. We would also like to thank Prof Eugene Katz for the help with designing the experiment and Dr Guy Ankonina for the help with optical measurements.

## Author contributions

C.R. and A.M. conceived the project. A.M. and C.R. developed the thermodynamic model. A.M. and N.K. performed the device simulations. A.M. performed the experiments. T.S. fabricated the glass samples. C.R. guided the research.

## Additional information

**Competing financial interests:** The authors declare no competing financial interests.

