## [Peer Review File · Nature Communications]

Note from the editor: this manuscript was reviewed previously at another journal.

Reviewer #2 (Remarks to the Author)

The work proposes a novel method for sunlight-to-electricity conversion with potential to exceed the Shockley-Queisser limit. A thermally insulated photoluminescent material is placed between the incident solar radiation and the PV cell. Concentrated sunlight excites the PL material and the thermalization losses bring the PL material up to operating temperature. The high temperature blue shifts the emission spectrum, thus by tuning the bandgaps of the PL material and PV cell energy that would ordinarily be lost as thermalization in the PV cell can be captured. The authors present a theoretical treatment and an experimental demonstration of the TEPL effect.

According to the analysis presented, the PL material needs an EQE of 100% and a photon-recycling efficiency of 98% to beat the SQ limit at 1100 K and 500 suns. The lowest EQE that can beat the SQ limit under those conditions is 91%, but the temperature needed is 2300 K. Can an EQE over 91% be achieved realistically? Can 98% optical recycling be achieved--the PV will have dead area from contacts and there may be loss in the concentrating optics?

The experiment demonstrated thermally induced up-conversion by pumping a preheated sample of Nd³⁺ doped glass with a 914 nm laser (below the bandgap of GaAs) and generating 815 nm radiation capable of exciting the GaAs cell. I do not think the experiment captures all the key elements of TEPL. The EQE of the PL material was only ~30% and the interaction with the incident light was modest. Additionally, the experiment does not seem to be truly steady state because the energy required to blue-shift the 914 nm laser came from the preheating of the sample.

I understand that the experiment is meant to serve as a proof-of-concept not a working prototype. Nevertheless, the large gap between the theoretical limit of 70% efficiency and the demonstrated 1.4% that is not addressed. Without additional analysis, it does not inspire confidence in the technology. What are the fundamental reasons the efficiency was so low? I suspect low EQE and low absorption, but are there other effects? How can these shortcomings be addressed? What is a realistic upper bound on the efficiency?

The manuscript is well written, but the figures could be improved. Figs. 1 and 2c are nearly identical. Figs. 2a and 5 serve a redundant purpose but could cause confusion because of the inconsistent geometry. Fig. 3a would be better with Amps as the unit on the y-axis. Figs.

Reviewer #5 (Remarks to the Author)

The manuscript has been transferred to Nature Communications, and the authors have responded to the reviews of the original Nature Photonics submission. Regarding my request to improve the discussion of rate equations in the SI, the authors have substantially improved that section. My question whether the proposed design would be an improvement over existing multiple-layer cells, the authors, in their response to reviewer 2, note that multi-junction PVs have achieved efficiencies of 46%. This is very high and seems hard to beat by the authors' design, but the authors say that the multi-junction PVs are rather complex and expensive. The EQE needed to achieve efficiencies above the SQ limit is said to be 91%, which, as the authors point out, is very demanding. Unfortunately, in their response the authors have not clarified my question or assumption that the thermalization of carriers, excited by non-monochromatic light above the bandgap, is not identical to the process of up-conversion, and that in the real device illuminated by sun light, there are similar amounts of down-conversion and up-conversion.

Having read all 5 reviews, it appears to me that all reviewers are skeptical (to various degrees) that the authors' design can eventually work as advertised. Nevertheless, I tend to accept the authors' argument that their design, if it can be made to work, could have practical improvements over existing (multi-junction) designs, and, as I said in my first review, I think the idea is interesting and important. Therefore I would like to keep my original recommendation in favor of publication (in Nature Communications).

Reviewer #6 (Remarks to the Author)

I think that at this point the authors answered my questions adequately well and I see no reason why this work should not be published

Reviewer #7 (Remarks to the Author)

Review of NCOMMS-16-04588

This is an interesting concept that relates to spectral conversion at elevated temperatures for PV applications. While the idea may have some merits, I am concerned about the following aspects, especially since this is intended for a Nature journal:

1) Using a 532nm laser as a source of "average" photons is not acceptable. A white light source should be used - e.g. quartz halogen lamp, xenon lamp, or even a supercontinuum laser - and this could be coupled into the system either via fibre optic cable (with collimator) or free-space optics as desired.

2) The value of the PL EQE is extremely important, however the authors do themselves a disservice by mis-quoting two references. The sentence "whereas Nd³⁺ doped glasses are reported to achieve PL EQE's of almost 100% 27,28" is not true. A quick read of these two references indicates that the >90% quantum yield values that are being reported are actually an INTERNAL value and not EXTERNAL, i.e. 90% of the ABSORBED photons are being re-emitted, whereas what the authors suggest is that this is actually an EXTERNAL quantum yield, i.e. that 90% of the INCIDENT photons are being re-emitted. For some material systems such as fluorescent organic dyes there is little difference between these two values, however the rare-earth ions exhibit such weak absorption that the external quantum yield is often several factors lower than the internal quantum yield due to these optical losses. Furthermore, the authors fail to point out that even the 90% internal quantum yield values were reported for extremely low Nd³⁺ doping concentrations (0.5%) - this is so low that this is never going to absorb any significant amount of light. To compound this problem, increasing the Nd³⁺ concentration, e.g. to 3.5% results in a drop in the internal quantum yield to 40-50%. Thus, this does not seem a promising approach.

3) Following on from point 2), I am confused as to how a Nd³⁺-doped glass layer can be referred to as having a bandgap of 1.1eV - either I have missed a major step or something is terribly wrong here. This is also an important link between the theory and experimental sections so this needs to be clarified.

4) Regarding cost issues, there are two options for the authors. Either A) stick to science and not enter a cost debate where there are many unknowns, or B) provide a detailed cost-analysis for the technology and highlight how this will be cheaper than the "expensive" 46% multi-junction solar cells, and calculate the cost of electricity. I think you will choose option A)...

Overall, until the above issues are adequately addressed I cannot recommend this manuscript to be published in Nature Communications.

Reviewer #2 (Remarks to the Author):

The work proposes a novel method for sunlight-to-electricity conversion with potential to exceed the Shockley-Queisser limit. A thermally insulated photoluminescent material is placed between the incident solar radiation and the PV cell. Concentrated sunlight excites the PL material and the thermalization losses bring the PL material up to operating temperature. The high temperature blue shifts the emission spectrum, thus by tuning the bandgaps of the PL material and PV cell energy that would ordinarily be lost as thermalization in the PV cell can be captured. The authors present a theoretical treatment and an experimental demonstration of the TEPL effect.

According to the analysis presented, the PL material needs an EQE of 100% and a photon-recycling efficiency of 98% to beat the SQ limit at 1100 K and 500 suns. The lowest EQE that can beat the SQ limit under those conditions is 91%, but the temperature needed is 2300 K. Can an EQE over 91% be achieved realistically? Can 98% optical recycling be achieved--the PV will have dead area from contacts and there may be loss in the concentrating optics?

We thank the referee for this question, indeed the EQE and the photon recycling are the most challenging parameters to realize. As a result of the reviewer's requests, we added a new white-light experiment, where the broad-band absorption is achieved by Cr:Yb:Nd:Glass. Such dopant materials have established values of EQE above 95% (See refs 21-23, 33-34).

A new simulation based on the white light experiment shows that EQE and PR efficiency of 90% each can support device efficiency above the SQ limit (40%) operating at 1500K. Higher EQE and PR of 95% can support device efficiency as high as 45%, also at 1500K. (See results on p. 13-15)

The simulation takes into account available state-of-the-art GaAs PV with R=98% of the back reflector, supporting PR~95% which matches an efficiency range of 40%-45%. For higher PR values a reflectivity above 98% is required, which can be achieved by state-of-the-art back-reflectors [Refs 35-39]

At this level we choose not to include any additional loss channel that may be further improved by PV and optical engineering (e.g. dead contact areas), which is very close to the SQ limit for a single junction cell.

The experiment demonstrated thermally induced up-conversion by pumping a preheated sample of Nd³⁺ doped glass with a 914 nm laser (below the bandgap of GaAs) and generating 815 nm radiation capable of exciting the GaAs cell. I do not think the experiment captures all the key elements of TEPL. The EQE of the PL material was only ~30% and the interaction with the incident light was modest. Additionally, the experiment does not seem to be truly steady state because the energy required to blue-shift the 914 nm laser came from the preheating of the sample.

I understand that the experiment is meant to serve as a proof-of-concept not a working prototype. Nevertheless, the large gap between the theoretical limit of 70% efficiency and the demonstrated 1.4% that is not addressed. Without additional analysis, it does not inspire confidence in the technology. What are the fundamental reasons the efficiency was so low? I suspect low EQE and low absorption, but are there other effects? How can these shortcomings be addressed? What is a realistic upper bound on the efficiency?

The manuscript is well written, but the figures could be improved. Figs. 1 and 2c are nearly identical. Figs. 2a and 5 serve a redundant purpose but could cause confusion because of the inconsistent geometry. Fig. 3a would be better with Amps as the unit on the y-axis. Figs.

We thank the referee for this comment. We agree that the experiment certainly does not capture all the key elements of TEPL. However, we stress that the up-conversion (UC) experiment was not a TEPL proof of concept – it is meant to give a quantitative result of the TEPL main feature, which is the thermal upconversion of sub-band photons by the thermalization energy of the hot photon absorption. As such, we compare it to existing UC methods and show its current efficiency, although not perfect, is in line with the best UC methods using monochromatic incoherent light.

Specifically, we choose to compare it to a thermal UC scheme which was published in Nat. Comm. 2014. This thermal method, which lacks the chemical potential counterpart of the emitted light, achieves 16% efficiency at 3000K but would only achieve 10^{-8} efficiency at 600K, which is our operating temperature. It is true that our sample's EQE is far from optimal, and better results could be obtained when optimizing the sample specifically for monochromatic UC aims. In the new experiment of white-light absorption we used different composition of PL materials and discuss ways to achieve high EQE.

Regarding the large gap between the theoretical limit of 70% efficiency and the demonstrated 1.4%: As mentioned above, we choose to compare the UC experiment with existing UC results. As for the 70% limit, the right comparison should be between the new white light experiment based device simulation (48%) and an ideal device (70%). Detailed analysis of the efficiency dependence on parameters such as the EQE and photon recycling efficiency can be found at the new discussion section (p. 14-15)

Regarding the steady-state issue: The idea behind the UC experiment is to show that the thermalization heat can induce efficient UC, which is at the heart of the TEPL mechanism. For this, the 532 pump must be switched off for the measurement of 914nm UC, since apart from the heating, the 532nm pump also induces its own PL - which cannot be present at the measurement. In the new experiment, there is no need to separate the two pumps since the white light drives both the PL and the thermalization, and therefore it's in true steady-state.

The manuscript is well written, but the figures could be improved.

We thank the reviewer for this useful comment; we have changed the figures in the text to better explain it without redundancy (Figure 1 was added an explanatory figure; figures 2 a,c were removed and replaced by a better device illustration; figure 3 which was quadruple was changed to a double-figure; figures 5 and 6 are new)

Reviewer #5 (Remarks to the Author):

The manuscript has been transferred to Nature Communications, and the authors have responded to the reviews of the original Nature Photonics submission. Regarding my request to improve the

discussion of rate equations in the SI, the authors have substantially improved that section. My question whether the proposed design would be an improvement over existing multiple-layer cells, the authors, in their response to reviewer 2, note that multi-junction PVs have achieved efficiencies of 46%. This is very high and seems hard to beat by the authors' design, but the authors say that the multi-junction PVs are rather complex and expensive. The EQE needed to achieve efficiencies above the SQ limit is said to be 91%, which, as the authors point out, is very demanding. Unfortunately, in their response the authors have not clarified my question or assumption that the thermalization of carriers, excited by non-monochromatic light above the bandgap, is not identical to the process of up-conversion, and that in the real device illuminated by sun light, there are similar amounts of down-conversion and up-conversion.

Having read all 5 reviews, it appears to me that all reviewers are skeptical (to various degrees) that the authors' design can eventually work as advertised. Nevertheless, I tend to accept the authors' argument that their design, if it can be made to work, could have practical improvements over existing (multi-junction) designs, and, as I said in my first review, I think the idea is interesting and important. Therefore I would like to keep my original recommendation in favor of publication (in Nature Communications).

Answer

We first would like to thank the reviewer for his positive opinion. Regarding the un-answered question:

“Unfortunately, in their response the authors have not clarified my question or assumption that the thermalization of carriers, excited by non-monochromatic light above the bandgap, is not identical to the process of up-conversion, and that in the real device illuminated by sun light, there are similar amounts of down-conversion and up-conversion”.

We are sorry for omitting this information from our answer – we missed it since it wasn't written as a direct question to us.

We shouldn't assume there are equal amount of up-and down conversions. When a bandgap absorber absorbs the solar light, the population of electrons and holes are, at the first instant, much hotter than the lattice. Upon cooling, much heat has to be dissipated for the carriers to cool down to 300K. This situation is characterized by the maximal chemical potential possible, and from there, any addition rise in temperature (due to insulation for example) would induce up-conversion or TEPL, on the expense of chemical potential decrease. Our detailed balance approach exactly calculates this spectral shift.

Reviewer #6 (Remarks to the Author):

I think that at this point the authors answered my questions adequately well and I see no reason why this work should not be published

Reviewer #7 (Remarks to the Author):

Review of NCOMMS-16-04588

This is an interesting concept that relates to spectral conversion at elevated temperatures for PV applications. While the idea may have some merits, I am concerned about the following aspects, especially since this is intended for a Nature journal:

1) Using a 532nm laser as a source of "average" photons is not acceptable. A white light source should be used - e.g. quartz halogen lamp, xenon lamp, or even a supercontinuum laser - and this could be coupled into the system either via fibre optic cable (with collimator) or free-space optics as desired.

We thank the reviewer for this important comment. We added new experiments in white-light TEPL conversion with a broadband absorber and a supercontinuum source. The new data is depicted at the end of the results section (p.10-12). We also added a new simulation of a practical white-light (Solar) device which is based on the experimental results showing expected efficiency of 47%-50%. We also added a discussion regarding the challenges in realizing such a device.

2) The value of the PL EQE is extremely important, however the authors do themselves a disservice by mis-quoting two references. The sentence "whereas Nd³⁺ doped glasses are reported to achieve PL EQE's of almost 100% 27,28" is not true. A quick read of these two references indicates that the >90% quantum yield values that are being reported are actually an INTERNAL value and not EXTERNAL, i.e. 90% of the ABSORBED photons are being re-emitted, whereas what the authors suggest is that this is actually an EXTERNAL quantum yield, i.e. that 90% of the INCIDENT photons are being re-emitted. For some material systems such as fluorescent organic dyes there is little difference between these two values, however the rare-earth ions exhibit such weak absorption that the external quantum yield is often several factors lower than the internal quantum yield due to these optical losses. Furthermore, the authors fail to point out the even the 90% internal quantum yield values were reported for extremely low Nd³⁺ doping concentrations (0.5%) - this is so low that this is never going to absorb any significant amount of light. To compound this problem, increasing the Nd³⁺ concentration, e.g. to 3.5% results in a drop in the internal quantum yield to 40-50%. Thus, this does not seem a promising approach.

We are sorry for confusing our definitions with the cited papers definitions. In the paper, we always assume full absorption of light, and for that reason, we treated the QE reported in the papers as EQE. Although the Reported doping concentration required to reach >90% QE are low, our

simulated device is based on measured experimental data of a Cr:Nd:Yb broadband absorber where the doping ratios are 0.1:1:0.4 (wt %), i. e. low concentrations. We take an advantage of the low absorption coefficient in the practical model, as we designed it for separation between the longitudinal dimension (in which the solar radiation is guided and absorbed) and the transverse, where PL is emitted towards the surrounding PV's. The longitudinal dimension is set to be sufficient for absorption of 98% of the solar pump. The new white light experiments and the practical device results can be found in pages 10-15 and in new figures 5-6.

3) Following on from point 2), I am confused as to how a Nd³⁺-doped glass layer can be referred to as having a bandgap of 1.1eV - either I have missed a major step or something is terribly wrong here. This is also an important link between the theory and experimental sections so this needs to be clarified.

This misunderstanding is our fault. In the prior revision, only the Nd³⁺ upconversion experiment existed and we tried to show the similarity between this experiment and the TEPL concept, which employs two bandgaps. This was an error since the rare-earths have energy-gaps with narrow absorption spectrum and cannot be treated as bandgap materials. The updated version corrects this approach with the new white-light demonstration, which utilizes a sensitized absorber to create a badgap-like, continuous absorption and emission spectrums. We corrected this issue in the text and in figure 4, in addition to the new white-light experiment and figures.

4) Regarding cost issues, there are two options for the authors. Either A) stick to science and not enter a cost debate where there are many unknowns, or B) provide a detailed cost-analysis for the technology and highlight how this will be cheaper than the "expensive" 46% multi-junction solar cells, and calculate the cost of electricity. I think you will choose option A)...

Thank you – we will stick with option A. we have taken off a comment regarding the multi-junction cost issues.

In general: due to the requested changes and additional experiments, we also changed the paper's layout so it first presents the conceptual model and theoretical results of the ideal thermodynamic model. We then proceed to the monochromatic up-conversion experiment (which already existed in the prior version), and link it to the white-light experiment (broadband upconversion). The new discussion section analyses the practical device with simulations and the main challenges before realization.

In addition, we edited and discarded a few of the figures in light of Reviewer 2 comments. We have changed the figures in the text to better explain it without redundancy (Figure 1 was added an explanatory figure; Figures 2 a,c were removed and replaced by a better device illustration; Figure 3 which was quadruple was changed to a double-figure; figures 5 and 6 are new).

Reviewer #2 (Remarks to the Author)

The authors have performed a white light experiment as recommended by multiple reviews. A custom fabricated Cr: Nd: Yb doped glass sample was illuminated first illuminated with three lasers then with a supercontinuum laser (and a CO₂ laser for additional heating). The TEPL up conversion peaks were shown for the three lasers. No spectrum was shown for the white light, but 5-30% (depending on temperature) of the emission was below 850 nm, indicating up conversion.

According to the authors, a unique property of TEPL is increase in the voltage but not the current of the PV cell. The first experiment with the GaAs PV shows a current increase, and the voltage increase in Fig 4b looks like it can be attributed solely to increased photocurrent. I suspect that strong optical coupling between the components is necessary for the voltage increase, possibly caused by the PL material interacting with the PV cell's radiative recombination. Although the experiment demonstrated up conversion, the optical coupling was not sufficiently strong to demonstrate this effect.

I am not convinced that the experiments sufficiently validate the theoretical framework that the authors use to predict high efficiencies. Furthermore, I remain unconvinced that TEPL will ever have an impact on solar energy because of the high optical efficiency and EQE required for efficiencies that are not significantly in excess of existing multijunction solar cells. These factors, combined with the uninspired figures, prevent me from recommending publication of the manuscript in its present form.

Reviewer #5 (Remarks to the Author)

I have read the authors response to my previous review, as well as the reviews (and responses) of reviewers #2, 6 and 7. Basically, this has reinforced my believe that the reviewers are skeptical (to various degrees) that the authors' design can eventually work as advertised. But, as before, I think the idea is interesting and important and sufficiently supported by the presented research, even if the eventual realizability is still in question, and therefore I recommend in favor of publication.

In this version, there have been substantial changes to the figures. I have to admit that I don't fully understand the physical processes that lead to figure 1a, but I assume I could obtain a better understanding if I would study reference 14 in detail. Here, I only want to point out that in the new version of figure 1, the authors use "A.U.". It is not clear whether that means "atomic units" or "arbitrary units" (I assume it means the latter). Moreover, it is confusing to me that Photon Rate is shown in the main part of figure 1a and in the inset. From what I understand, the inset shows a spectrally integrated value (from 1.45eV to infinity), whereas the main part shows a spectral density. In other words, I would have thought these are different quantities with different units, and hence should have different names. The fact that the units are omitted makes it very difficult to understand properly what is shown in the figure. This is also a problem in figure 4b. I suggest that this be improved. On a positive note, I like the new version of figure 2.

Reviewer #7 (Remarks to the Author)

Overall, the authors have put in a solid effort to address the reviewers' comments. Yes, there are still several short-comings however for a proof-of-concept I believe that enough has been done and that the paper can now be accepted to Nature Comms.

Reviewer 2:

“The authors have performed a white light experiment as recommended by multiple reviews. A custom fabricated Cr:Nd:Yb doped glass sample was illuminated first illuminated with three lasers then with a supercontinuum laser (and a CO2 laser for additional heating). The TEPL up conversion peaks were shown for the three lasers. **No spectrum was shown for the white light**, but 5-30% (depending on temperature) of the emission was below 850 nm, indicating up conversion. “

The requested spectrum was presented in figure 5b (inset) but it wasn't made clear enough that this spectrum is a result of white-light excitation. We have now put it in a new panel (Figure 6c) and added the appropriate explanatory text in the figure. In addition, the white excitation source spectrum (Fianium) is depicted in Supplementary Figure 2.

“According to the authors, a unique property of TEPL is increase in the voltage but not the current of the PV cell. The first experiment with the GaAs PV shows a current increase, and the voltage increase in Fig 4b looks like it can be attributed solely to increased photocurrent. I suspect that strong optical coupling between the components is necessary for the voltage increase, possibly caused by the PL material interacting with the PV cell's radiative recombination. Although the experiment demonstrated up conversion, the optical coupling was not sufficiently strong to demonstrate this effect.”

As stated in the text, the experiment presented in figure 4b (Figure 5a in the new version) is an up-conversion experiment, which demonstrates key feature of the TEPL where sub bandgap photons are up-converted by temperature. This results in enhancement of the current at the GaAs cell, which also leads to voltage enhancement. The reviewer is correct in principle that strong optical coupling is contributing for the voltage enhancement effect in the TEPL model, but this is not part of this demonstration.

Reviewer 5:

“I only want to point out that in the new version of figure 1, the authors use "A.U.". It is not clear whether that means "atomic units" or "arbitrary units" (I assume it means the latter). Moreover, it is confusing to me that Photon Rate is shown in the main part of figure 1a and in the inset. From what I understand, the inset shows a spectrally integrated value (from 1.45eV to infinity), whereas the main part shows a spectral density. In other words, I would have thought these are different quantities with different units, and hence should have different names. The fact that the units are omitted makes it very difficult to understand properly what is shown in the figure. This is also a problem in figure 4b. I suggest that this be improved. On a positive note, I like the new version of figure 2.”

We have changed the Y-axis annotation in figure 1b to “Energetic photon rate” in order to differentiate it from figure 1a and figure 4c.